# Neural Activation in Risky Decision-Making Tasks in Healthy Older Adults: A Meta-Analysis of fMRI Data

**DOI:** 10.3390/brainsci11081043

**Published:** 2021-08-06

**Authors:** Thomas Tannou, Eloi Magnin, Alexandre Comte, Régis Aubry, Sven Joubert

**Affiliations:** 1Laboratoire de Recherches Intégratives en Neurosciences et Psychologie Cognitive—UR LINC, UFC, UBFC, 25000 Besançon, France; eloi.magnin@univ-fcomte.fr (E.M.); alexandre.comte@univ-fcomte.fr (A.C.); raubry@chu-besancon.fr (R.A.); 2Inserm, CIC 1431, Centre d’Investigation Clinique, University Hospital of Besançon, 25000 Besançon, France; 3Geriatrics Department, University Hospital of Besançon, 25000 Besançon, France; 4Centre de Recherche Institut Universitaire de Gériatrie de Montréal, Université de Montréal, Montréal, QC H3W 1W6, Canada; sven.joubert@umontreal.ca; 5Neuraxess Functional Neuroimaging and Neurostimulation Platform, University Hospital of Besançon, 25000 Besançon, France; 6Neurology Department, University Hospital of Besançon, 25000 Besançon, France; 7Département de Psychologie, Université de Montréal, Montréal, QC H2V 2S9, Canada

**Keywords:** aging, decision making, fMRI, meta-analysis

## Abstract

Decision making is a complex cognitive phenomenon commonly used in everyday life. Studies have shown differences in behavioral strategies in risky decision-making tasks over the course of aging. The development of functional neuroimaging has gradually allowed the exploration of the neurofunctional bases of these behaviors. The purpose of our study was to carry out a meta-analysis on the neural networks underlying risky decision making in healthy older adults. Following the PRISMA guidelines, we systematically searched for fMRI studies of decision making in older adults using risky decision-making tasks. To perform the quantitative meta-analysis, we used the revised version of the activation likelihood estimation (ALE) algorithm. A total of 620 references were selected for initial screening. Among these, five studies with a total of 98 cognitively normal older participants (mean age: 69.5 years) were included. The meta-analysis yielded two clusters. Main activations were found in the right insula, bilateral dorsolateral prefrontal cortex (dlPFC) and left orbitofrontal cortex (OFC). Despite the limited number of studies included, our meta-analysis highlights the crucial involvement of circuits associated with both emotion regulation and the decision to act. However, in contrast to the literature on young adults, our results indicate a different pattern of hemispheric lateralization in older participants. These activations can be used as a minimum pattern of activation in the risky decision-making tasks of healthy older subjects.

## 1. Introduction

Decision making is a complex cognitive phenomenon that is important and commonly used in everyday life situations. Because older people are at increased risk of developing severe pathologies that impact their functional independence, they have to face more decision-making situations [1]. These decisions concern, for example, consent to a specific treatment. However, it can also involve complex choices, such as moving from their home to a facility better adapted to their health condition. This type of decision making is a dynamic process that involves a balance between risk-taking, the preservation of independence and safety. However, through accumulated experience, older people also have the ability to use a range of experiences to support their decision making [2,3,4].

Decision making refers to a process that includes several steps, from analyzing a problem, to taking action to solve it. In the medical field, the ability to make a decision relies on the ability to provide informed consent [5,6,7]. Decision making has been broken down into three cognitive stages: (i) option generation, (ii) option evaluation, and (iii) launching the action [8,9]. In addition to mobilizing the sensory functions necessary for communication and language skills involved in the integration of information, decision making involves complex neuropsychological processes. Indeed, decision making is largely part of executive functioning and involves other processes such as flexibility, inhibition, working memory and emotion recognition.

Neuropsychological tests that assess risky decision making typically use two types of conditions: (1) decisions “under ambiguity”, where the probability associated with each outcome is unknown, and (2) decisions “under risk”, where the rule of attribution and occurrence of an event is known [10]. Risk preferences often differ depending on whether people make choices based on described probabilities, versus direct experience of the odds and outcomes in behavioral tasks [11]. In decisions based on experience, which are closest to the real-life decision, the most frequently used behavioral test is the Iowa Gambling Task (IGT), which is generally recognized as a decision-under-ambiguity task [12]. Other related behavioral tests include the Balloon Analogue Risk Task (BART) [13] and the Game of Dice Task (GDT) [14]. Computerized versions of those behavioral tasks have been adapted and are widely used in the literature. Nevertheless, the classification allowing the distinction between “under ambiguity” and “under risk” is based on the main period of interest analyzed. Indeed, in the IGT and BART, the principle is that there are classically two phases in these tasks, associated with a learning curve. Initially, the participant explores strategies corresponding to “under-ambiguity” experimentation. In a second phase, the participant has understood which strategies are advantageous or not, switching to a model called “under-risk”. However, the BART involves much more risk taking and impulsivity. Indeed, the participant has direct feedback on his/her strategy (explosion or gain) and can adapt his/her risk-taking accordingly for the following trials [13]. However, in the IGT, the participant tries to understand the winning strategy, which is not explicit [15]. In the GDT, on the other hand, the consequences of choices are explicitly given in advance. Consequently, the GDT is an exclusive decision-making task under risk, without learning by feedback. These tests have been used in numerous studies in both healthy young and older populations, as well as in clinical populations (with psychiatric disorders, in particular) [16,17,18]. These studies have shown qualitative and quantitative differences in behavioral strategies in these tasks as a function of age, the presence of neurological lesions and/or neuropsychiatric pathologies.

There is evidence that decision-making processes can be affected over the course of normal aging, especially when decisions must be taken quickly, and that this may be due to an underlying decline in processing speed [19,20]. A decline in executive functioning in aging such as increased perseverative behavior may also be associated with risky decision-making impairment. Thus, when faced with new and unfamiliar situations, older adults take more time to analyze the situation than younger ones. This has been observed during decision-making tests in situations where the outcomes and consequences are unknown, such as the first phase of the IGT or BART [21]. On the other hand, in a previous meta-analysis, it was noted that, in behaviorally risky decision-making tasks, the need for learning had an impact on the choice strategy in aging compared to younger subjects [22]. As soon as the rules become clear, i.e., in the “under-risk” final phases of behavioral tasks, however, this difference between younger and older healthy adults disappears. Thus, when older adults are given enough time or when they can rely on past experiences, decision-making function remains preserved pp. 351–370 in [1,23]. Furthermore, the decision-making processes involved in these tests include executive functions, but also motivation and reward processes. In general, tests tend to show that aging is associated with a change in strategy; a reduction or prevention of losses is favored over the optimization of gains. Choices resulting from this shift are influenced by the individual motivation profile in each situation. These data are also found in ecological and neuro-economic analyses [24].

The development of functional neuroimaging has gradually allowed the exploration of the neurofunctional bases of these behaviors. However, studies on decision making have been carried out mainly in healthy young adults. In this specific population, several meta-analyses have been performed [25,26,27]. Results have mainly shown activation of the right dorsolateral prefrontal cortex (dlPFC) [27], especially in sub-risk decision-making situations (which are most similar to complex real-life decision-making situations), followed by the activation of the orbitofrontal cortex (OFC) and the insula [26]. In recent years, studies comparing young versus older healthy participants have shown inconsistent results in terms of patterns of activation, which may depend on the types of tasks used. As such, healthy older subjects are reported to have greater contralateral activation in prefrontal and insular regions compared to young adults [28,29], while other authors have described an age-related increase in right ventro-medial prefrontal cortex activity [29,30]. Nevertheless, although the use of neuroimaging techniques to study decision-making abilities associated with aging is growing rapidly, there are many fewer studies with this population than with younger adults, and the results are still sometimes ambiguous and difficult to interpret.

Thus, it seemed important at this stage to determine, based on these data, if there was a common pattern of functional activation related to experience-based decision making in risky situations in normal aging emerging from these studies. The purpose of our study was to carry out a meta-analysis on the neural networks underlying risky decision making in healthy older adults.

## 2. Materials and Methods

This meta-analysis was conducted following the guidelines from the Preferred Reporting Items for Systematic Reviews and Meta-Analyses (PRISMA) [31]. The details of the method are explained below:

### 2.1. Criteria for Inclusion

Participants: We included only studies that included healthy older participants. To target an aging population, and to ensure that we did not miss any relevant studies, we included studies with a minimal inclusion age of 55 years. This is typically the minimum age for the inclusion of older participants.

fMRI: Functional magnetic resonance imaging or functional MRI (fMRI) measures brain activity by detecting changes associated with blood flow, which creates what is referred to as a blood-oxygen-level-dependent (BOLD) signal. Thus, Voxel-Based Morphometry analyses in anatomical MRI were not considered. Diffusion and resting state data were also not included.

Decision-making tasks in risky situations: This corresponds to both “sub-ambiguity” and “under risk” behavioral tasks, as well as mixed decision-making neuropsychological tests. As such, we narrowed our inclusions to widely used and replicated tasks such as the Balloon Analogue Risk Task (BART), the Game of Dice Task (GDT) and the Iowa Gambling task (IGT).

### 2.2. Search Strategy

We searched for publications specifically evaluating decision making in older adults. Databases of peer-reviewed literature were systematically searched on PubMed and EMBASE for manuscripts in the English language published up to December 2020.

The primary search term was as follows: (aged or older or elder or elderly or geriatrics or senior or adults or adult) AND (fMRI or “functional magnetic resonance imaging” or “functional neuroimaging” or “functional MRI”) AND (“Decision-making” or “Decision making”) AND (“Iowa Gambling Task” OR IGT or “Game of Dice Task” or GDT OR “Balloon Analogue Risk task” OR BART).

This search yielded 620 articles. The titles and abstracts were reviewed to refine the number of potentially relevant articles. All the articles in which the abstract did not specifically mention the age of the participants, or the nature of neuropsychological tests (risk-based decision-making test), were fully reviewed. Following this step, the inclusion criteria were assessed for each study by carefully reading the Methods and Results sections of each article.

As a final step, if an article met the search criteria, at this stage, all the references were examined one by one by the authors and the articles were screened using “Connectedpaper.com” algorithms, to make sure that no study had been omitted.

Within the selected articles, the data extracted included the age of the subjects, level of education, global cognitive assessment, type of decision-making test and coordinates of hyperactivation peaks in the decision-making phase. Among the data analyzed in fMRI, as we were interested in the decision-making phase and not reward-based neurological processes, data associated with reward times were excluded.

### 2.3. Statistical Analysis

All the studies included in the systematic review were also assessed for a meta-analysis after a quality check with the Covidence Quality Assessment Template. To be included in the meta-analysis, studies had to report (i) specific peaks of foci of activation in either Talairach or Montreal Neurological Institute (MNI) space and (ii) peak activation coordinates of whole-brain analyses (could not employ only region of interest, ROI, analyses).

To perform quantitative meta-analysis, we used the revised version of the activation likelihood estimation (ALE) algorithm [32,33] which treats foci as 3-dimensional Gaussian probability distributions centered at the given coordinates.

Based on these recommendations, to correct for multiple comparisons, a cluster-level family-wise error (cFWE) at *p* < 0.05 was applied using uncorrected *p* < 0.001 at the voxel level as the cluster-forming threshold (based on 1000 permutations).

To perform ALE meta-analysis, we used specialized software (GingerALE v.3.2.0) to combine the activation coordinates from several studies [34]. All the peak voxel coordinates were reported in MNI space. For consistency, peak voxels reported in Talairach space in primary studies were converted into MNI space, and the ALE results were displayed onto the MNI brain template using the Mango software package.

The performance of a meta-analysis requires some vigilance in the case of small numbers of included studies [35]. In our study, we distinguish significant results as clusters from non-significant peaks. To avoid biases, only cluster results will be detailed and discussed.

## 3. Results

### 3.1. Study Selection

A total of 620 references were selected for initial screening, and, among them, 136 duplicates were removed. As such, the titles and abstracts of 484 studies were screened. After this first screening, 60 studies were assessed for full-text eligibility. Finally, after careful analyses, 55 studies were excluded. The main reasons for exclusion were as follows: 37 were excluded for “non-target population” (subjects younger than our criteria), 15 for non-target study design (no risky decision-making tests) and three for non-target outcome (ROI analysis and connectivity). As a result, only five studies fully satisfied all the criteria including passing quality checking and were included for meta-analysis [30,36,37,38,39]. A flow chart is shown in Figure 1.

These five studies included a total of 98 healthy subjects aged from 58 to 95, with a mean age 69.5 years, and were cognitively normal. The behavioral tests used were the BART in one publication, the GDT in one other and the IGT in the three remaining studies. All the study characteristics and behavioral results are summarized in Table 1. Some studies included only healthy older adults, another compared healthy and Parkinson Disease older adults and another compared younger and older adults. The five studies came from different teams.

Concerning the paradigms used, for GDT, Labbuda and al. [36] used an adapted paradigm with no feedback phase. For the BART task, Yu et al. [37] used a two-level adaptation of the Lejuez paradigm [13], with low and high risk using two different colors, but without explaining it to participants. Concerning the three IGT studies included, they are all based on the Bechara initial task adapted by Li for fMRI sessions [40]. As such, the three paradigms are quite similar, with a four-choice situation. Halfmann and al. [38] reduced the number of trials to 80.

Concerning the fMRI analysis, only based on the decision phase, our systematic review yielded 22 foci to be analyzed from the 98 participants.

### 3.2. Main Clusters

The meta-analysis yielded two clusters, with four peaks in the first cluster and two peaks in the second one. The full results are detailed in Table 2 and Figure 2A,B.

The first cluster is 16.752 mm^3^, from (12, 0, −16) to (58, 42, 34), centered at (38.4, 21.4, 5.3) with four peaks and a max value of 0.0079 ALE, 1.8139492 × 10^−4^ P, 3.57 Z at (38, 20, −3). It is located in the right brain hemisphere, with 76.3% sub-lobar activation and 23.7% Frontal Lobe activation. Anatomically, this corresponds to the Lentiform Nucleus (49%), the Insula (17.4%), the Inferior Frontal Gyrus (11.4%), the Middle Frontal Gyrus (10.1%), the Claustrum (4.8%), the Caudate (3.5%) and the Precentral Gyrus (2.6%). In terms of functional areas, these are mainly the Putamen (38.8%), Brodmann area 13 (17.1%), Brodmann area 9 (9.2%), the Lateral Globus Pallidus (8.8%), Brodmann area 47 (5.4%), the Caudate Head (3.5%), Brodmann area 45 (3.5%), Brodmann area 46 (2.5%), Brodmann area 44 (2.1%) and the Medial Globus Pallidus (1.4%).

This corresponds to activations of the right dorsolateral prefrontal cortex (dlPFC), inferior frontal gyrus (IFG) and cortico-basal loops of the motivation and reward circuits including the insula and putamen.

The second cluster is 11.048 mm^3^, from (−44, 10, −24) to (−14, 48, 8), centered at (−27.6, 30.1, −9.4) with two peaks and a max value of 0.0114 ALE, 4.2548763 × 10^−6^ P, 4.45 Z at (−22, 34, −14). It is located in the left-brain hemisphere with 55% sub-lobar activation and 45% Frontal Lobe activation. Anatomically, this corresponds to the Insula (36.7%), the Inferior Frontal Gyrus (30.6%), the Claustrum (15.8%), the Middle Frontal Gyrus (14%) and Extra-Nuclear (1.8%). In terms of functional areas, these are mainly Brodmann area 13 (33.8%), Brodmann area 47 (25.5%), Brodmann area 11 (12.6%) and Brodmann area 45 (5.8%).

This corresponds to activations of the left orbitofrontal cortex (OFC), the left dlPFC and cortico-basal loops of the motivation and reward circuits.

## 4. Discussion

Our meta-analysis of patterns of neural activation during behavioral risky decision making in healthy older adults showed clusters of activation in the dorsolateral prefrontal and orbitofrontal cortex, as well as in cortico-basal loops. Despite an inclusion criterion >55 years old, the mean age of the study population was 69.5 years with few participants from 55 to 60 and many participants older than 60, up to 95 years old Thus, our study concerned older participants.

The results of the current meta-analysis are in line with previous findings showing involvement of both dlPFC and cortico-basal loops to support executive functioning and cognitive automatization, as well as involvement of the orbitofrontal cortex and the insulae to support emotion and reward loops. In this regard, these findings are similar to what has been reported in younger healthy adults [28].

Indeed, the insula is involved in emotional regulation circuits, particularly in relation to reward circuits and disgust. The activation of the insula could be associated with reduced risk taking and with more conservative behavior, potentially associated with risk avoidance [41].

Concerning the inferior frontal gyrus, recent studies have highlighted its importance in working memory and cognitive flexibility in decision making [42]. 

In our meta-analysis, these activations were found in the bilateral orbitofrontal cortex, although predominantly in the left hemisphere, classically known for its involvement in the modulation of decision making in reaction to the identification of emotional processes with projections within the insula [43]. In addition, bilateral activation (although predominantly right) was found in the putamen and the dorsolateral prefrontal cortex, associated with executive functions, cognitive automatization and the planning of the response in relation to the decision taken [44].

In contrast to young adults, however, results of the current meta-analysis indicate that older adults show a different pattern of functional activation in several regions. In fact, we observed a bilateral, although predominantly right, pattern of activation of the cortico-basal loops. Although our study population does not allow direct comparison with younger subjects, the literature shows that this pattern of activation seems to be predominantly left-sided in young subjects [26].

Various functional neuroimaging studies in cognitive aging have led to the development of models, such as the Hemispheric Asymmetry Reduction in Older Adults (HAROLD) [45,46] and the Posterior-Anterior Shift in Aging (PASA) theory, trying to explain these modifications seen in aging [47]. The HAROLD model predicts that aging is associated with the recruitment of additional neural resources in regions contralateral to those seen in younger people in the prefrontal cortex [45,48]. Even though ROI structural imaging studies on risky decision making in aging seem to support the HAROLD model [28], we were unable to find evidence to confirm any of those models in our meta-analysis. Indeed, the results of the current meta-analysis rather seem to support the view that the pattern of functional activation is contralateral to that found in young adults, rather than bilateral, at least with respect to decision making.

The set of behavioral tasks included in our meta-analysis are risk-taking decision-making tasks. However, among the five included studies, four used tasks combining decisions under ambiguity and decisions under risk, while the fifth task required only decisions under risk. Although all of these tasks mobilize executive functions, we must remain careful in not over-interpreting the results of our meta-analysis, since the tasks used cognitive processes to varying degrees, namely, impulsivity, learning by feedback and risk taking. This limits the generalizability of our results and warrants the need for further studies on complex decision making to be carried out, especially within the context of aging.

Thus, this question of compensation in aging with possible recruitment of additional areas still remains partially unexplained in decision making. In this respect, our meta-analysis is not powerful enough and other studies need to be conducted to be able to better analyze this type of compensation. Although our study was limited in its statistical power, it approached this question in an original way and synthesized the current known data. To be able to really interpret the neurofunctional activation data, a greater homogeneity of the paradigms and protocols of decision-making tasks and fMRI analysis must be sought. Although the paradigms included in our study are not identical, the behavioral approach to risky decision making is similar enough across tasks to warrant analysis. However, paradigm adaptations to age-related vulnerabilities, including simplifications of tasks, are observed, sometimes deviating considerably from the initial behavioral task. On the other hand, the same acquisition, analyzed by several teams, can lead to divergent results [43]. This necessitates a quality process for future research. Given the lack of standardization, we chose to restrict our meta-analysis to widely used tasks that have been the subject of behavioral analyses reproduced in the literature, to avoid biases related to tasks that would not respect risky decision-making behaviors. This leads our analysis to focus on a small number of studies but favors the comparability of the studies and thus the relevance of the results. It is widely accepted that the more the studies included in a meta-analysis, the better the usability of the results [49]. However, because of the narrowness of our inclusion criteria and the exploitation of only the results grouped in the form of clusters, without integrating the isolated peaks, we reduced the risk of bias [50]. We therefore present our results as a minimum activation pattern. Nevertheless, these results should be interpreted with caution and confirmed by further studies as the functional research on decision making in older adults grows.

Thus, our meta-analysis describes a model of activation in the left OFC, bilateral dlPFC and right insula that can be used as a minimum pattern of activation in the risky decision-making behavioral tasks of healthy older subjects.

## 5. Conclusions

In conclusion, our meta-analysis on the neural correlates of risky decision making in normal aging highlights the involvement of circuits associated with both emotion regulation (insula, inferior frontal gyrus and orbito-frontal cortex) and executive function (Putamen and dorsolateral prefrontal cortex). 

## Figures and Tables

**Figure 1 brainsci-11-01043-f001:**
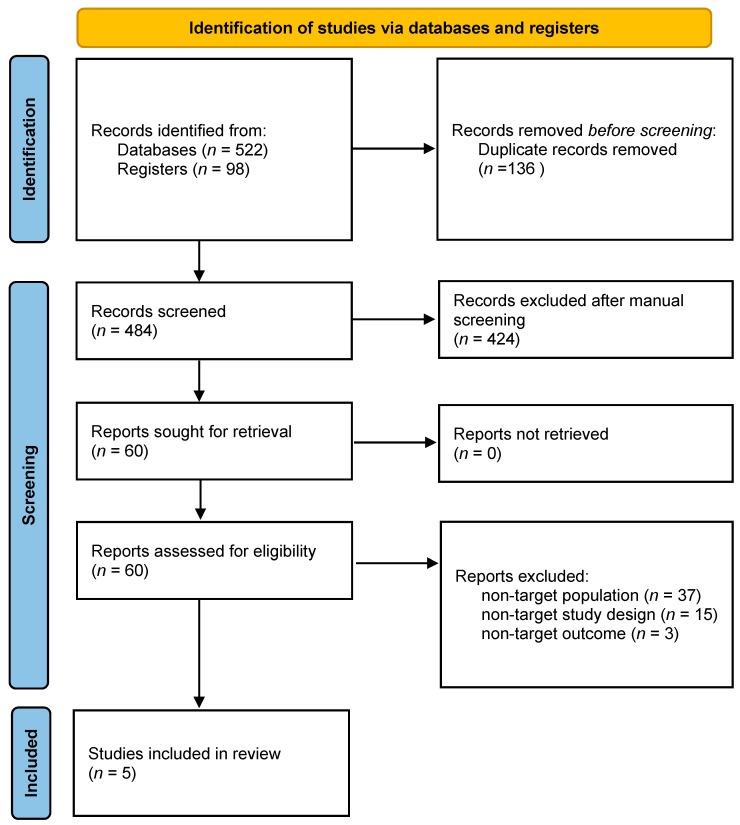
PRISMA Flow chart.

**Figure 2 brainsci-11-01043-f002:**
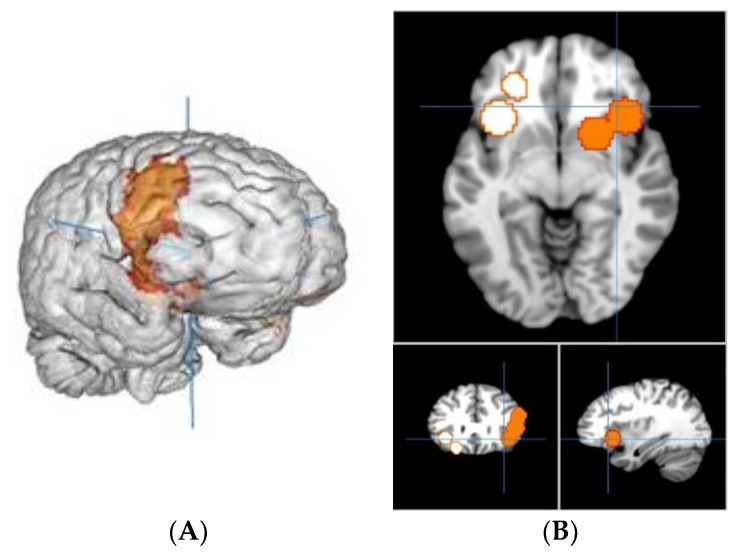
(**A**) Right view of cluster 1; (**B**) slice view of cluster 1 in orange and cluster 2 in white.

**Table 1 brainsci-11-01043-t001:** Characteristics of the 5 included studies.

Study	Year	Age (Mean) or (Range)	Number of Participants	Education (Years)	Cognitive Score	Behavioral Test	Performance on Decision-Making Test
Labudda et al. [36]	2008	62.33 (4.81)	12	10.92 (4.81)	DemTect >13/18 Mean: 16.83 (1.64)	GDT	Advantageous alternative: 91.29%
Yu et al. [37]	2016	65.3 (5.3)	23	*NA*	MMSE >26 Mean: 29.1 (1.2)	BART	Proportion of cashout trials: 70%
Rogalsky et al. [30]	2012	77 (58–95)	14	*NA*	MMSE >26 Mean: 29.46	IGT	Advantageous score: 2.272 Disadvantageous: −2.16
Halfmann et al. [38]	2014	77 (59–88)	31	16.2 (3.0)	Full Scale IQ Mean: 120.8 (10.9)	IGT	Advantageous score: 6.5 Disadvantageous: 1.5
Paz et al. [39]	2019	63 (9.7)	18	14.97 (3.29)	Premorbid IQ 50 (44.8–57.3)	IGT	no risk–risk, %: 38.9 (29.9)

**Table 2 brainsci-11-01043-t002:** Clusters and peaks.

Cluster #	x	y	z	ALE	P	Z	Area	Broadman	Hemisphere
1	38	20	−3	0.007913446	1.8139492 × 10^−4^	3.5657701	Insula	13	R
	46	26	10	0.0076704714	3.5643848 × 10^−4^	3.3845782	Inferior Frontal Gyrus	45	
	21	10	−7	0.0076653096	3.5643848 × 10^−4^	3.3845782	Putamen		
	50	32	24	0.007145069	4.9175817 × 10^−4^	3.2952	dlPFC	9	
2	−22	34	−14	0.011441755	4.2548763 × 10^−6^	4.451949	Orbito-frontal Cortex	47	L
	−34	20	−2	0.007665314	3.5643848 × 10^−4^	3.3845782	Insula		
Peaks non-significant as clusters	18	−90	−4	0.008416306	8.338653 × 10^−5^	3.7646651	Lingual Gyrus	18	R
	6	−90	−14	0.0071455813	4.9175817 × 10^−4^	3.2952	Declive (cerebellum)		
	2	28	36	0.0071442807	5.164076 × 10^−4^	3.2814317	Cingulate Gyrus	32	L
	8	38	14	0.007555881	3.6959953 × 10^−4^	3.3746119	Anterior Cingulate	32	R
	−34	10	46	0.0071442807	5.164076 × 10^−4^	3.2814317	Middle Frontal Gyrus	6	L
	−48	−58	42	0.0071442807	5.164076 × 10^−4^	3.2814317	Inferior Parietal Lobule	40	L
	15	−13	−4	0.0076653096	3.5643848 × 10^−4^	3.3845782	Thalamus		R
	57	−27	−12	0.008135796	1.40038 × 10^−4^	3.633064	Middle Temporal Gyrus	21	R
	48	−45	51	0.008135796	1.40038 × 10^−4^	3.633064	Inferior Parietal Lobule	40	R
	−14	−61	17	0.008617656	7.734643 × 10^−5^	3.7834134	Posterior Cingulate	30	L
	−4	60	9	0.0089207785	5.4598368 × 10^−5^	3.8691933	dlPFC	9	L
	−6	0	51	0.008410512	9.895277 × 10^−5^	3.7216759	Cingulate Gyrus	24	L
	−4	8	−17	0.0089207785	5.4598368 × 10^−5^	3.8691933	Subcallosal Gyrus	25	L
	4	−74	54	0.0071442807	5.164076 × 10^−4^	3.2814317	Precuneus	7	R
	−34	−84	30	0.009234563	2.4616009 × 10^−5^	4.059246	Superior Occipital Gyrus	19	L

## Data Availability

All the data can be made available upon reasonable request to the corresponding author.

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
