# Peer review of "Neural Activation in Risky Decision-Making Tasks in Healthy Older Adults: A Meta-Analysis of fMRI Data"

_brainsci, 2021, doi:10.3390/brainsci11081043_

Round 1
Reviewer 1 Report
The paper reports a meta-analysis of fMRI studies of decision-making in healthy older adults using risky decision-making tasks. Results showed that, in contrast to the literature on young adults, older adults reported a different pattern of activation. Particularly, the Authors found the involvement of circuits associated with emotion regulation and decision to act (right insula, bilateral dorsolateral prefrontal cortex, and left orbitofrontal cortex).
This is a relevant topic in the area of the decision-making process in aging and this manuscript will provide a relevant contribution to the field.
The strength of this study is that it is the first metanalysis investigating this issue. The limit regards the small number of studies included.
Overall, I think that the Authors make a good job in describing the literature and discussing results. There are only a couple of issues that need to be addressed before publication:
- In the introduction (lines 61-62), the Authors described the differences between decisions under ambiguity and under risk. However, it is not clear the categorization of decision tasks. For example, the IGT is a decision under ambiguity task, while the Game of Dice Task is usually categorized as a decision under risk task. The Authors should better describe this section.
- Criteria for inclusion (lines 117 – 118) about the age of 55 years can be questionable. Older adult age is classified starting from 65 years old. The Authors should explain the reason to include studies with older adults 55-64 years old.
Minor points:
- Line 39-41: Please rephrase the sentence.
- Please pay attention in using decision-making or decision making in the manuscript using the same criteria.
- Line 129: “Analogue” instead of “Analog”
- Line 196: “Bechara” instead of “Bacchara”
Author Response
Dear reviewer,
Thank you for your evaluation of our article. Your careful study has revealed points of improvement that improve the overall quality of our manuscript.
Thus:
- regarding the lack of clarity in the description (introduction) of the tasks, we modified the paragraph by adding the following elements in red within sentences:
“In decisions from experience, that are closest to the real-life decision, the most frequently used behavioral test is the Iowa Gambling Task (IGT) which is generally recognized as a decision under ambiguity task (12). Other related behavioral tests include the Wisconsin Card Sorting Task (WCST) (13), the Balloon Analogue Risk Task (BART) (14) and the Game of Dice Task (GDT) (15) which are generally recognized as decision under risk tasks. Computerized versions of those behavioral tasks have been adapted and are widely used in the literature. Nevertheless, the classification allowing the distinction between "under ambiguity" and "under risk" is based on the main period of interest analyzed. Indeed, the principle is that there are classically two phases in these tasks, associated with a learning curve. Initially, the participant explores strategies, corresponding to "under-ambiguity" experimentation. In a second phase, the participant has understood which strategies are advantageous or not, switching to a model called "under-risk".”
- Regarding the age criteria, in method section, we have modified the paragraph by added the following sentence:
We included only studies that included healthy older participants. To target an aging population, and to ensure that we did not miss any relevant studies, we included studies with a minimal inclusion age of 55 years. This is typically the minimum age for inclusion of older participants.
We also reinforced results with :
"Despite an inclusion criterion > 55 years old, the mean age of the study population was 69.5 years with few participants from 55 to 60, and many participants older than 60, up to 95 y.o. Thus, our study concerned older participants."
We also modified the minor points you pointed out.
Reviewer 2 Report
In the present paper, the authors carried out an ALE meta-analysis on neuroimaging data in order to highlight neural activations associated with risky decision-making in healthy adults.
Unfortunately, it was possible to include in the meta-analysis only 5 works. This greatly affects the reliability of the results. As recommended by the guidelines for meta-analysis (Müller et al., 2018), the studies to include must be more, otherwise the generalizability of effects is questionable. "Based on a recent simulation study (Eickhoff et al., 2016), a recommendation was made to include at least 17–20 experiments in ALE meta-analyses in order to have sufficient power to detect smaller effects and to also make sure that results are not driven by single experiments."
This seems to be particularly important in this case, where very different tasks are included in the same analysis (e.g., IGT; BART and GDT).
As it is now, the meta-analysis has not enough works to ensure that the meta-analysis has adequate sensitivity to detect effects of the expected magnitude, while maximizing ability to generalize to as broad a population of studies of interest as possible.
Muller, V.I., Cieslik, E.C., Laird, A.R., Fox, P.T., Radua, J., Mataix-Cols, D., …, Eickhoff, S.B., 2018. Ten simple rules for neuroimaging meta-analysis. Neurosci. Biobehav. Rev. 84, 151–161. doi: 10.1016/j.neubiorev.2017.11.012 .
Author Response
Dear reviewer,
We thank you for the careful reading of our manuscript and the comments you have suggested, linked with the small number of studies included.
Your remarks led us to complete both the methodological section and, above all, the discussion, and to had a sentence in the abstract.
Indeed, we agree on the benefit of including many studies to reinforce the significance of the results of meta-analyses, and the criterion set by Müller et al. is one perspective on this question. Nevertheless, as mentioned by Dr. Michael Borrenstein, this does not prevent meta-analyses from being performed with fewer studies, as long as (1) the inclusion and analysis criteria are sufficiently narrow to reduce heterogeneity of results, and (2) there is no overgeneralization.
To secure compliance with these elements, we have strengthened the following elements in our manuscript:
- First, the included paradigms are quite similar. As specified in "Method":
As such, we narrowed our inclusions to widely used and replicated tasks such as the Balloon Analogue Risk Task (BART), the Game of Dice Task (GDT, Wisconsin Card Sorting Task (WCST) and the Iowa Gambling task (IGT).
- On the other hand, we did not exploit individual peaks, but only clusters, which avoids over-representation of a single study:
"The performance of a meta-analysis requires some vigilance in case of small numbers of included studies (36). In our study, we distinguish significant results as clusters from non-significant peaks. To avoid biases, only cluster results will be detailed and discussed."
- Finally, the discussion of the limitations associated with this small number of studies has been reinforced. In addition to the elements already mentioned, i.e. the precaution on the range of selected tasks, and using only a cluster analysis without taking into account isolated peaks, we sought to determine a minimum pattern of activation. Further studies will certainly add to our work, which must therefore be interpreted with caution.
It is widely accepted that the more studies included in a meta-analysis, the better the usability of the results (50). However, because of the narrowness of our inclusion criteria and the exploitation of only the results grouped in the form of clusters, without integrating the isolated peaks, we reduce the risk of bias (51). We therefore present our results as a minimum activation pattern. Nevertheless, these results should be interpreted with caution, and confirmed by further studies as the functional research on decision making in the older adults grows.
In the abstract, we added : "Despite the limited number of studies included, our meta-analysis highlights ..."
--
36. Valentine JC, Pigott TD, Rothstein HR. How Many Studies Do You Need?: A Primer on Statistical Power for Meta-Analysis. J Educ Behav Stat. 2010;35:215‑47.
50. Müller VI, Cieslik EC, Laird AR, Fox PT, Radua J, Mataix-Cols D, et al. Ten simple rules for neuroimaging meta-analysis. Neurosci Biobehav Rev. 2018;84:151‑61.
51. Borenstein M, Hedges LV, Higgins JP, Rothstein HR. Introduction to meta-analysis. John Wiley & Sons; 2021.
Round 2
Reviewer 2 Report
I am persuaded by the explanations raised by the authors. No further comment.
Author Response
Thank you for your feedback. Your comments have improved the quality of the manuscript.